# Study on Treatment of Tiny Pollution Water with PAC-HUM System in Kuitun River

**DOI:** 10.3390/membranes12101010

**Published:** 2022-10-18

**Authors:** Liang Pei, Jia Duo

**Affiliations:** 1National Engineering Technology Research Center for Desert-Oasis Ecological Construction, Xinjiang Institute of Ecology and Geography, Chinese Academy of Sciences, Urumqi 830011, China; 2Xinjiang Key Laboratory of Environmental Pollution and Bioremediation, Xinjiang Institute of Ecology and Geography, Chinese Academy of Sciences, Urumqi 830011, China; 3Institute of Geographic Sciences and Natural Resources Research, Chinese Academy of Sciences, Beijing 100101, China; 4University of Chinese Academy of Sciences, Beijing 100049, China

**Keywords:** imitated tiny pollution water, powder activated carbon (PAC), flux, hollow fiber

## Abstract

Kuitun city, Xinjiang is dry and short of water, so it is urgent to treat and utilize all kinds of unconventional water. In view of this problem, we conducted a study on the treatment of tiny pollution water in Kuitun River. We investigated the effect of dosage of powder activated carbon (PAC) on hollow-fiber ultrafiltration membrane (HUM) performance. The results show that the stable operation time of hollow fiber ultrafiltration membranes lengthened and the rate of reduction of the flux was reduced when the PAC dosage was increased. The addition of PAC had no obvious effect on the resistance of membrane filtration. We conducted experiments to evaluate the effect of ultrafiltration of tiny pollution water in combination with PAC. When the parameters of operation and PAC dosage were appropriately regulated, the removal rates of chemical oxygen demand (COD), ammonia nitrogen (NH_3_-N) and ferric ions (Fe) reached 62%, 32% and 90%, respectively. When the PAC dosage was 200 mg/L, 100 mg/L and 150 mg/L, the highest removal rates were achieved under normal temperature and pressure. The effluent COD was less than 5.0 mg/L, NH_3_-N was less than 1.5 mg/L and Fe was less than 0.5 mg/L, achieving better results than the quality standard of surface water (GB3838-2002). The treated water can be discharged into the river or recirculated to utilities. The fouled membrane was cleaned by water rinsing, water/acid rinsing and water/alkali rinsing, with recovery ratios of 44%, 81% and 88%, respectively. The results of this study can serve as a foundation for the efficient utilization of water resources and the development of new water treatment technologies in Xinjiang.

## 1. Introduction

Water is a precious natural resource, and the effective treatment of polluted water has attracted widespread attention. Traditional water treatment technology is difficult to adapt to modern changes in water quality [1,2]. Xinjiang is an arid area with serious water shortage and backward water treatment and utilization technology. Kuitun city is an overexploited area of groundwater, and water resources are extremely scarce. The Kuitun River is a slightly polluted river in northern Xinjiang. It is necessary to recycle water after treatment [3,4]. Xinjiang is an underdeveloped area in China with limited economic conditions. The development of water treatment technology is still relatively preliminary in this area. The methods used in Kuitun are mainly traditional oxidation ditches and simple wetland treatment methods. The treated water quality cannot meet the requirements of local water resource recycling, and the membrane method has not been applied locally due to various restrictions. Therefore, it is necessary to popularize and apply membranes for water treatment and membrane technology in Xinjiang. The results of this study can serve as a foundation for the efficient utilization of water resources and the development of new water treatment technologies in Xinjiang [5,6].

Compared with conventional processes, the UF process has the following advantages: ① A high turbidity removal rate, as well as stable and reliable effluent quality. In terms of turbidity and particulate matter removal, the UF process achieves a higher removal rate than conventional processes. The effluent turbidity of the UF process is stable below 0.1 NTU, and the removal rate of particulate matter can reach 99.9% [7,8,9]. ② The UF process can effectively remove pathogenic microorganisms, such as Cryptosporidium, bacteria and other pathogenic microorganisms and viruses in water [7,8,9]. The ultrafiltration effluent does not need to be disinfected, and there are no byproducts of chlorination disinfection, although disinfectant left in the water itself has an impact on human health. ③ The water plant covers a small area. The UF process only requires about 1/5 the floor are of traditional processes [10]. ④ Compared with conventional and advanced treatment processes, the ultrafiltration process saves water production cost.

The external engineering example shows that the cost of water production is basically the same as that of conventional processes. Based on statistical analysis, Klaus [11] concluded that the water production cost of the UF process in Germany was DEM 0.095/m^3^ (about CNY 0.55/m^3^; the membrane renewal cost is equivalent to 5 years of service life; the electricity cost is calculated as 0.15 kw/m^3^ of electricity consumption). Fang [12] used an integrated water purification device to treat water from the Yangtze River water with a pretreatment + ultrafiltration process, resulting in a large water yield. When the inflow is 16.83–17.67 L/min, the outflow is 15.00–16.40 L/min; the outflow is equivalent to 89.0–93.0% of the inflow, with good effluent quality. The whole unit covers an area of about 1.28 m^2^, which saves a considerable area compared with traditional processes. Wu et al. [13] found that the water production cost of UF in China is CNY 0.23/m^3^, and the power consumption is 0.18 kw/m^3^. Research by the Shenzhen Water Group shows that adding an ozone-activated carbon advanced treatment process to a conventional process increases the cost of water production by CNY 0.30/m^3^. The abovementioned studies show that the water production cost of the ultrafiltration process is lower than that of a conventional process + ozone + activated carbon advanced treatment process. P. Lipp et al. [8] compared the investment cost of the UF process and with that of conventional processes based on French engineering practice and concluded that with a 2 × 10^4^ m^3^/d small-scale water plant, UF has a lower investment cost than conventional technology. In the case of a medium-scale (2 × 10^4^~10 × 10^4^ m^3^/d) water plant, there is little difference between the investment cost of UF and that of conventional processes. For water plants larger than 30 × 10^4^ m^3^/d, the investment cost of UF is slightly higher than that of conventional processes. The water production cost of the UF process mainly consists of membrane renewal fees, chemical agent consumption, equipment maintenance fees and electricity fees. Membrane renewal costs account for the largest proportion (about 42%), followed by electricity costs (32%), pharmaceutical costs (21%) and maintenance costs (5%). Once the ultrafiltration process is widely used in the field of water treatment, the price of ultrafiltration membranes is bound to continue to decline, in addition to the operating cost of ultrafiltration processes improving the competitive advantage of UF for water production [14].

In recent years, ultrafiltration (UF) technology has been widely used in the field of water treatment technology due to its characteristics of high-molecular-weight cutoff, low operating pressure and low operating costs [7,8,9]. However, it is difficult to meet the requirements for natural and synthetic organic substances or soluble small-molecule substances with microfiltration technology alone, which needs to be combined with other methods. Activated carbon is a porous material with a large specific surface area and high adsorption capacity. It is widely used for the adsorption and decolorization of chemical products, as well as wastewater treatment [8,9,10]. Many studies have shown that combining the adsorption of activated carbon with the interception of ultrafiltration membranes can effectively improve the performance of the treatment process and the removal rate of pollutants and slow down the decline rate of membrane flux. The combination of PAC and HUM is a research hotspot in water treatment. The advantage of this process is that it combines the adsorption of PAC on low-molecular-weight organics with the screening of UF on macromolecular organics, bacteria and other pathogenic microorganisms, considerably improving the removal rate of organics and effectively slowing down membrane pollution. The combined process of powdered activated carbon ultrafiltration (PAC-HUM) for drinking water treatment has been extensively studied by many scholars, who believe that it is a membrane process with the potential to replace conventional water treatment processes to effectively remove organic matters. Rama [15] proposed the addition of powdered activated carbon into the circulating water flow of a UF membrane device to form an adsorption solid–liquid separation process to treat drinking water. PAC can effectively adsorb low-molecular-weight organics in water, transfer dissolved organics to a solid phase and remove low-molecular-weight organics from water using a UF membrane to intercept and remove particles. Moreover, PAC can effectively prevent membrane fouling. Using electron microscopy, Joseph et al. [16] found that PAC formed a porous membrane layer on the membrane surface, which adsorbed organic substances in water, not only removing organic substances but also avoiding membrane pollution. This layer of PAC film is relatively soft and can be easily removed by backwashing. Research by Wang et al. [17] and Pei et al. [18] on the combined use of powdered activated carbon and ultrafiltration membranes to remove pollutants in drinking water shows that an increased concentration of powdered activated carbon, the removal effect of the permanganate index, UV254 and phenol is increased. The addition of PAC can enhance the ability of ultrafiltration membranes to remove permanganate index and NH_3_-N, but the efficiency and law of UV254 removal need to be further studied. Furthermore, the addition of powdered activated carbon can reduce membrane fouling and play an important role in maintaining the high specific flow rate of ultrafiltration membranes. Dong et al. [19,20] conducted experiments on the treatment of raw water from the Huangpu River with the combined process of powdered activated carbon and an ultrafiltration membrane and found that activated carbon and ultrafiltration membranes jointly undertake to remove organic matter in water, with complementary effects. For organic substances with a low molecular weight, the adsorption effect of activated carbon is better than that of membrane filtration. For organic substances with a high molecular weight, the membrane filtration effect is satisfactory, whereas the adsorption effect of activated carbon is poor. Many water plants have applied the combined PAC-HUM process, the most typical of which is Vigneux Water Plant in France, which was put into operation in 1997, with a scale of 5.5 × 10^4^ m^3^/d and an average PAC dosage of 8 mg/L [7]. At present, there are dozens of large-scale drinking water treatment plants in Europe using the combined process of PAC and UF membranes, with a total treatment capacity of 200,000 m^3^/d. The abovementioned studies and applications show that the combination of PAC adsorption and UF interception can improve the performance of the treatment process, slow down the decline rate of membrane flux and effectively reduce the formation of disinfection byproducts in water [21].

Despite the many applications and studies on ultrafiltration and PAC, these processes have not been promoted and applied in many poor areas in China, especially in the arid areas of northwest China, such as Xinjian, where we applied PAC-HUM for the first time. Kuitun River is a major river in northern Xinjiang, where large areas of farmland need to be irrigated by river water [22,23,24,25,26,27]. The current traditional water treatment technology and simple constructed wetland technology cannot treat water to irrigation and discharge standards. Due to economic constraints, the use of cheap membrane materials is suitable for local conditions. We used a hollow fiber ultrafiltration membrane, cheap PAN material and locally available PAC for this study. In this study, polyacrylonitrile (PAN) was used as hollow fiber membrane (HUM) material, and powder activated carbon (PAC) was used to carry out experimental research on micropolluted water treatment. The effect of PAC on membrane flux and the removal effect of the PAC-HUM system on COD, NH_3_-N and Fe^3+^ were investigated, providing a technical basis for further research on industrial applications. The results of this study can serve as a foundation for the efficient utilization of water resources and the development of new water treatment technologies in Xinjiang.

## 2. Materials and Methods

### 2.1. Experimental Materials

The experimental device and process flow are shown in Figure 1. The membrane module is a domestic hollow-fiber ultrafiltration membrane with a molecular weight cut-off (MWCO) of 100,000. The material is polyacrylonitrile, which a type of hydrophilic membrane. The pressure method is internal pressure, the membrane pore size is 0.03 µm and the effective surface area of the membrane is 3.55 m^2^. See Table 1 for performance indices of powdered activated carbon (PAC).

### 2.2. Experimental Methods

The raw water used in the experiment is slightly polluted water collected from the Kuitun River in late September. See Table 2 for water quality indicators. The hollow-fiber ultrafiltration membrane module adopts the operation mode of cross-flow operation, and the cross-flow speed is 0.15 m/s. The original mixed liquid is sent to the ultrafiltration membrane by a multistage centrifugal pump at a constant pressure (0.06 mPa), and the constant water inflow of the system is controlled by a regulating valve with pH = 6–9. The raw water first enters the complete mixing water tank (CSTR) and is fully mixed with the added PAC under the action of the agitator. The mixed liquid enters the membrane module through the pressure pump for separation. The concentrated liquid is drained into the water tank, and the leachate enters the outlet tank. The membrane module should be recoiled regularly (about every 5–8 days).

PAC is fed in batches. At the beginning of the experimental cycle, PAC was added to the reactor once. When PAC failed or the quality of effluent water did not meet the requirements, the PAC mixture was discharged, and new PAC was added.

The normal operating pressure of ultrafiltration equipment is 0.06–0.12 MPa (60–120 on the instrument display of the control cabinet). When the operating pressure of the ultrafiltration equipment reaches 0.12 MPa (the instrument in the control cabinet displays 120), forced-dosing backwashing is required.

## 3. Results and Discussion

### 3.1. Effect of PAC on Membrane Flux

As shown in Figure 2, when the same amount of water is treated, the decline in membrane flux after adding PAC is much smaller than that without adding PAC because PAC adsorbs some organic substances, thus reducing the membrane blockage caused by the adsorption and accumulation of pollutants in the water on the membrane surface. This shows that adding PAC can slow down membrane pollution. In addition, with increased PAC dosage, the amount of adsorbed pollutants increases, which reduces the chance of pollutants in the water blocking the membrane pores, in addition to loosening the mud cake formed by the large-particle disaster membrane surface with PAC as the core, thus better maintaining the membrane flux. Therefore, with a high PAC dosage, the decline in membrane flux is minimal [28,29,30,31].

### 3.2. Removal of COD_Mn_

As shown in Figure 3, Figure 4 and Figure 5, the removal rate of COD_Mn_ by HUM alone can only be maintained at about 20%, and after PAC is added, the removal rate can reach more than 30%. When the dosage is increased to 200 mg/L, the removal rate can reach more than 62% because when the PAC dosage is low, the concentration of PAC in the mixed liquid is low, and the total adsorption area is smaller than when the dosage is high. Therefore, PAC preferentially adsorbs low-molecular-weight organics, which may be most easily adsorbed by PAC. With increased PAC dosage, the total adsorption area also increases, and PAC starts to adsorb organic substances high molecular weights. In addition, the organic matter adsorbed by PAC during long-term operation becomes a hotbed for microbial propagation and forms a biofilm, which also promotes the removal of organic matter [25,26]. Furthermore, increased operation time and hydraulic retention time, PAC form a compacted layer on the membrane surface after adsorbing organic matter. The interface between the compacted layer and the membrane also forms a concentration polarization layer due to pressure. In this way, the pore diameter of the membrane is equivalent to continuously reducing, removing substances with an ionic radius that is smaller than the membrane pores, increasing the removal efficiency. However, as compacted layer and concentration polarization layer increasing, the flux also decreases, and the filtration pressure increases, resulting in the need to clean and restore the membrane regularly [21,28].

The COD_Mn_ of the effluent is about 5.0 mg/L, which is better than the environmental quality standard for surface water (GB3838-2002) [32].

### 3.3. Removal of NH_3_-N

As shown in Figure 6, the effect of HUM alone on NH_3_-N removal is not obvious, and the removal rate is about 5%. However, when PAC and HUM are combined, the removal rate of NH_3_-N increases because PAC forms a compacted layer on the HUM, which reduces the pore diameter of the membrane. As shown in Figure 7, the removal of NH_3_-N is divided into two stages: in the first stage, the removal rate of NH_3_-N is less than 10% at the initial stage of operation of the combined PAC-HUM process; after the system has operated for a period of time, the removal rate of NH_3_-N increases to 30%. In the second stage, after the membrane is backwashed, the NH_3_-N removal rate decreases, and as the experiment continues, the NH3-N removal rate starts to increase again, demonstrating that in the initial stage of operation until the membrane is backwash, pollutants in the raw water and PAC form a compacted gel layer on the membrane surface and inside the membrane pores, narrowing the membrane pore size and increasing the NH_3_-N removal rate. After backwashing, the gel layer falls off, increasing the pore size and decreasing the removal rate [23,24].

As shown in Figure 8, when the dosage of PAC is 20–50 mg/L, the NH_3_-N removal rate is low, and when the dosage is increased to 100 mg/L~250 mg/L, the removal rate reaches more than 32%. The NH_3_-N of the test effluent is less than 2.5 mg/L, which is superior to the environmental quality standard for surface water (GB3838-2002). However, the NH_3_-N removal rate is still low, indicating the need for further research.

### 3.4. Removal of Fe Ion

Figure 9 and Figure 10 are diagrams showing the removal effect of HUM alone and the combined PAC-HUM process on total iron, respectively. When the total iron content of the influent fluctuates considerably, the iron removal rate of the HUM system is between 58 and 78%, and the Fe content in the effluent is less than 0.5 mg/L, which is superior to the environmental quality standard for surface water (GB3838-2002). However, the iron removal rate of the combined PAC-HUM process is between 73 and 94.7%, which is significantly better than that of the HUM system alone because with increased operation time and hydraulic retention time, PAC form a compacted layer on the membrane surface after absorbing various pollutants. The interface between the compacted layer and the membrane also forms a concentration polarization layer due to pressure. In this way, the pore diameter of the membrane is equivalent to continuously reducing, removing substances with an ionic radius smaller than the membrane pores, resulting in increased removal efficiency. The radius of heavy metal ions is reduced in the process of operation, compared with the pore diameter of the membrane, which is constantly decreasing. However, with the increase in the compacted layer and concentration polarization layer, the flux will also decrease and the filtration pressure will increase, which requires us to clean and restore the membrane regularly [19,20,21,22].

It can be seen from Figure 11 that when the dosage of PAC is 20–100 mg/L, the Fe removal rate is only about 90%, while when the dosage of PAC is increased to 150 mg/L, the Fe removal rate is more than 90%.

### 3.5. Removal of SS, Coliform and Turbidity

Suspended solids SS and coliform bacteria have not been detected in the effluent during the whole experiment, and both of them have a 100% removal rate. HUM can completely intercept them. The turbidity of the effluent is below 0.1 NTU. The removal of these substances is mainly related to the physical interception of membrane pore size. It is also related to the increase in compaction layer formed by PAC and pollutants [20,21].

### 3.6. Membrane Pollution and Cleaning

With the increase in operation time and hydraulic retention time, PAC will form a compacted layer on the membrane surface after adsorbing pollution matters. The interface between the compacted layer and the membrane will also form a concentration polarization layer due to pressure. In this way, the pore diameter of the membrane is equivalent to continuously reducing, removing the substances whose ionic radius is smaller than the membrane pore, and the removal efficiency is getting higher and higher. However, with the increase in the compacted layer and concentration polarization layer, the flux will also decrease, and the filtration pressure will increase, resulting in the need to clean and restore the membrane regularly.

The normal operating pressure of ultrafiltration equipment is 0.06–0.12 MPa (60–120 on the instrument display of the control cabinet). When the operating pressure of the ultrafiltration equipment reaches 0.12 MPa (the instrument in the control cabinet displays 120), forced-dosing backwashing is required. After a period of operation of the combined PAC-HUM process, the membrane permeation pressure gradually increases to 0.12 Mpa, and the membrane flux gradually decreases to close to 0, indicating that the membrane has been seriously polluted. Water washing, water/pickling and water/alkali washing are performed, the membrane flux is measured after each cleaning. Water washing involves backwashing the water-permeable side with tap water until there is basically no pollutant flowing out. Water/pickling refers to soaking the membrane in 0.5% HC1 solution for 1.0 h and then backwashing with water. Water/alkali washing refers to washing the membrane with water backwashing method and soaking it 1% NaOH solution for 1 h, followed by water backwashing. As shown in Figure 12, the three cleaning methods can restore the membrane flux to 44%, 81% and 88% of that of a new membrane, respectively. Because the pollution of river water mainly originates from the illegal discharge of domestic sewage, which is mainly acidic, compound backwashing with alkaline liquid can react with acidic pollutants to remove the pollutants adsorbed on the membrane, which has the best recovery effect on the membrane [21,22,23]. Cleaning the membrane regularly can effectively remove membrane pollution, increase flux and reduce energy consumption.

## 4. Conclusions

(1)The experimental results show that adding PAC to an HUM system is an effective way to reduce membrane filtration resistance and improve membrane flux.(2)The removal rates of COD, NH_3_-N and Fe in wastewater by the combined PAC-HUM process can reach 62%, 32% and 90%, respectively. The best dosage of PAC to achieve a high removal rate is 200 mg/L, 100 mg/L and 150 mg/L for COD, NH_3_-N and Fe, respectively.(3)The effluent COD of the combined PAC-HUM process system was about 5.0 mg/L, that of NH_3_-N was less than 1.5 mg/L, that of Fe was less than 0.5 mg/L, suspended solids (SS) and coliform group were not detected, turbidity was below 0.1 NTU and the water quality was better than the environmental quality standard for surface water (GB3838-2002). The removal effect of organic matter, ammonia nitrogen and iron by PAC combined with HUM was better than that by HUM alone.(4)For contaminated membranes, water backwashing, water/acid washing and water/alkali washing can restore the membrane flux to 44%, 81% and 88% of that of a new membrane, respectively.

The results of this study provide a technical basis for further research on industrial applications, serving as a foundation for the efficient utilization of water resources and the development of new water treatment technologies in Xinjiang.

## Figures and Tables

**Figure 1 membranes-12-01010-f001:**
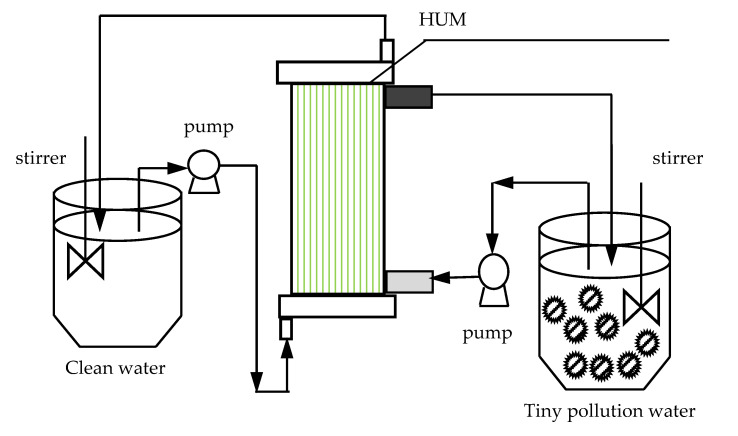
Process flow diagram of PAC-HUM.

**Figure 2 membranes-12-01010-f002:**
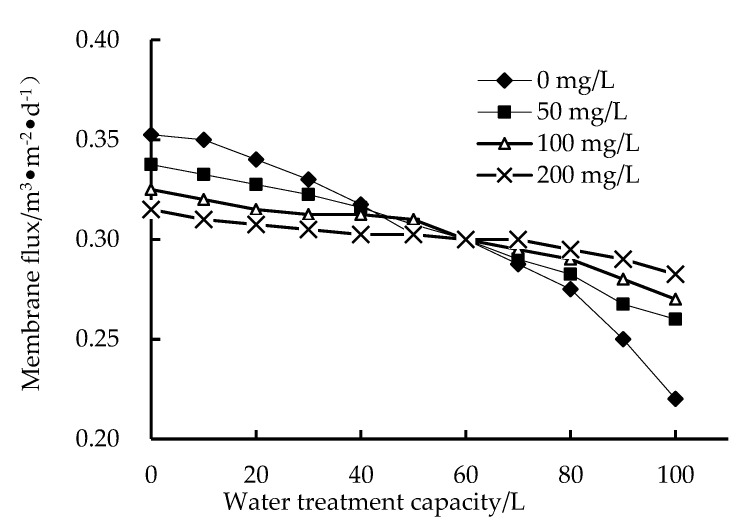
Effect of PAC dosage on the membrane.

**Figure 3 membranes-12-01010-f003:**
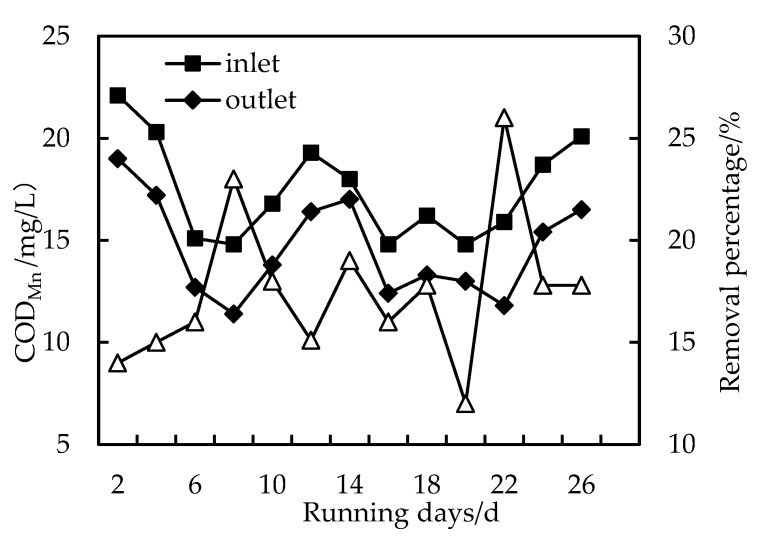
Removal of COD_Mn_ with HUM.

**Figure 4 membranes-12-01010-f004:**
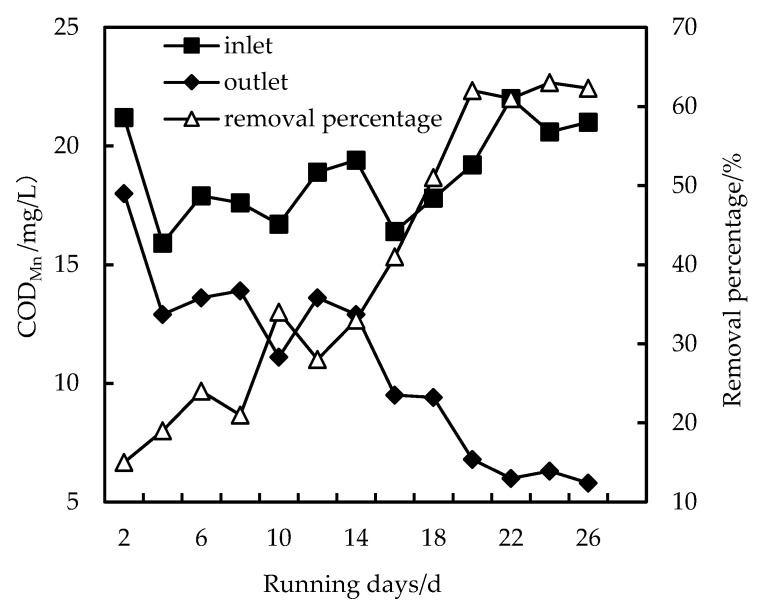
Removal of COD_Mn_ with PAC-HUM.

**Figure 5 membranes-12-01010-f005:**
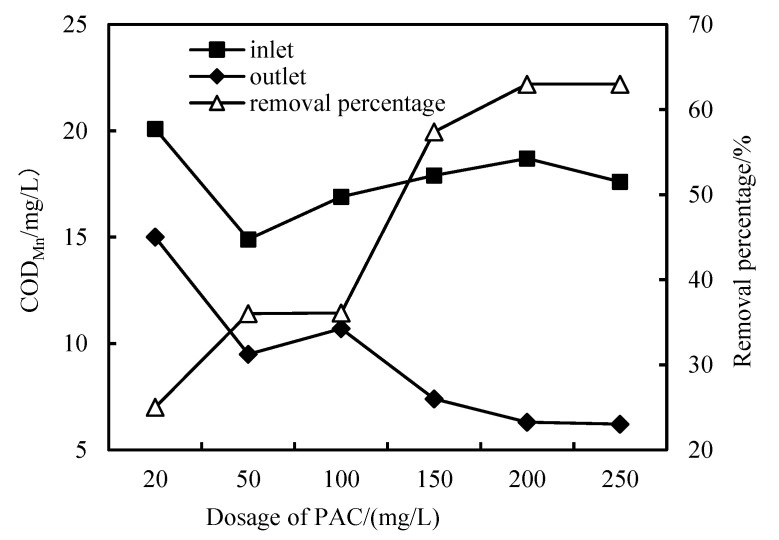
Removal effect of COD_Mn_ with PAC dosage.

**Figure 6 membranes-12-01010-f006:**
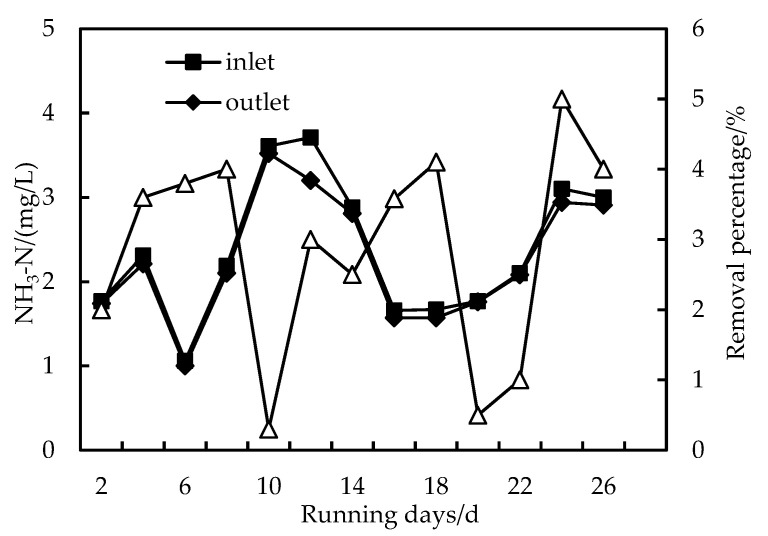
Removal of NH_3_-N with HUM.

**Figure 7 membranes-12-01010-f007:**
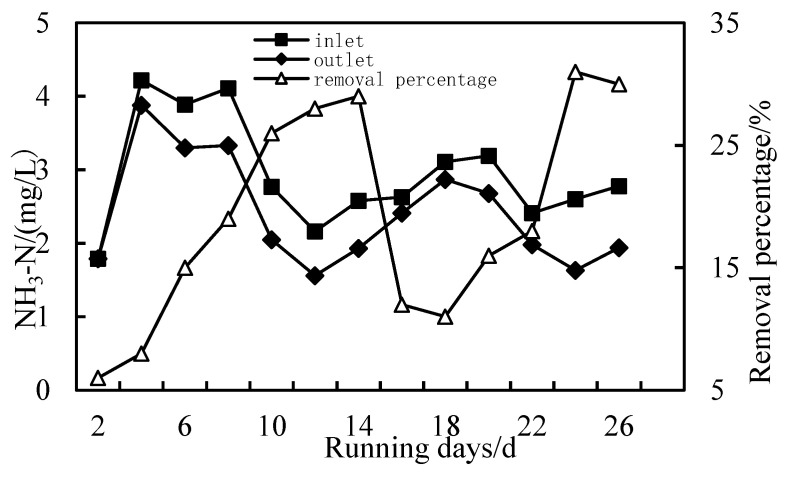
Removal of NH_3_-N with PAC-HUM.

**Figure 8 membranes-12-01010-f008:**
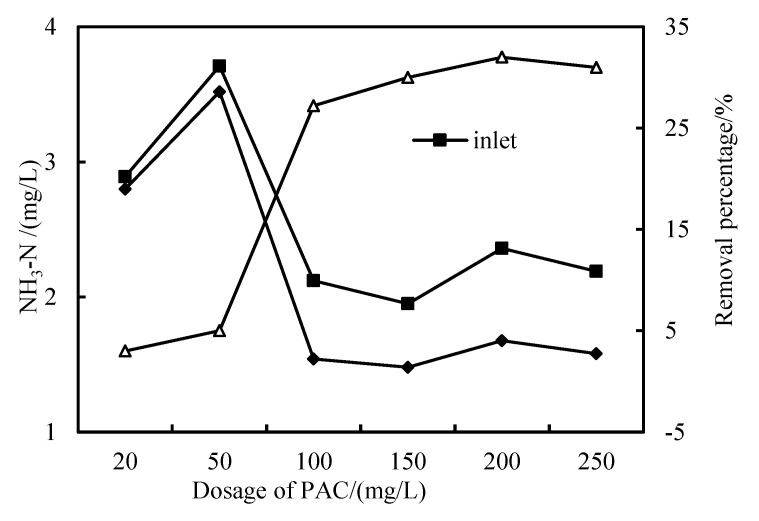
Removal effect of NH_3_-N with PAC dosage.

**Figure 9 membranes-12-01010-f009:**
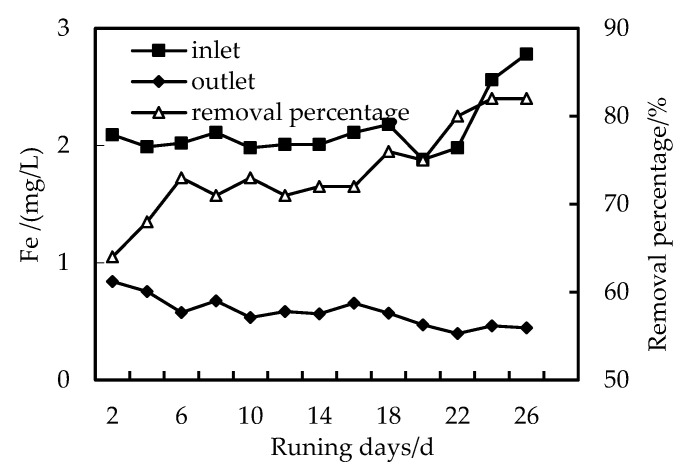
Removal of Fe with HUM.

**Figure 10 membranes-12-01010-f010:**
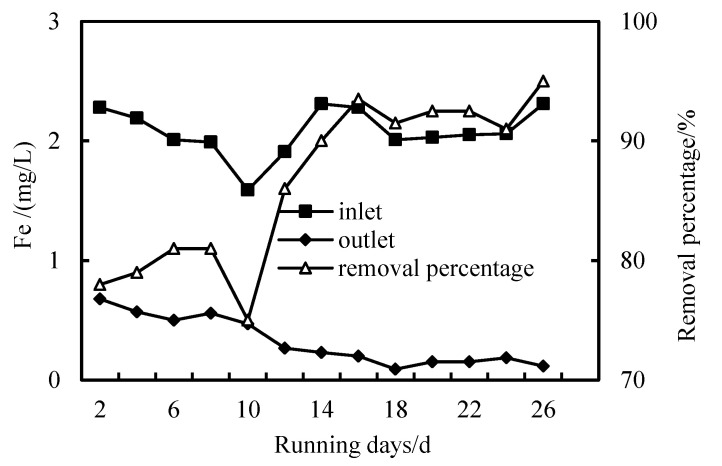
Removal of Fe with PAC-HUM.

**Figure 11 membranes-12-01010-f011:**
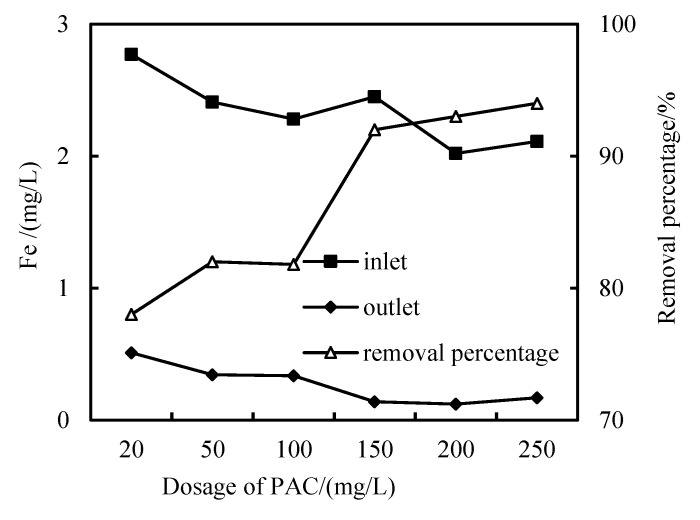
Removal effect of Fe with PAC dosage.

**Figure 12 membranes-12-01010-f012:**
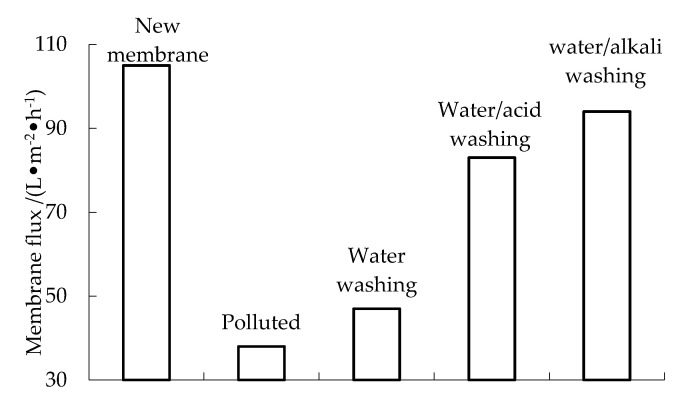
Recovery of membrane flux.

**Table 1 membranes-12-01010-t001:** Properties of PAC (type: ICT200).

Item	Index
True density/(kg/m^3^)	2.7 × 1000
Bulk density/(kg/m^3^)	0.69 × 1000
Iodine value/(mg/g)	979
Granularity/%	94.8 (200 mesh sieve)
Particle size/µm	20~400
Ash content/%	10.8
Water content/%	2.3

**Table 2 membranes-12-01010-t002:** Quality index of experimental raw water.

Item	COD/(mg/L)	TN/(mg/L)	NH_3_-N/(mg/L)	Turbidity/NTU	Fe/(mg/L)
Index	18.1~23.8	3.11~7.15	2.29~5.01	2.12~3.99	2.04~4.07

## Data Availability

Not applicable.

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
