# Peer review of "Study on Treatment of Tiny Pollution Water with PAC-HUM System in Kuitun River"

_membranes, 2022, doi:10.3390/membranes12101010_

Round 1

Reviewer 1 Report

There are several comments to be addressed by the authors:

Abstract:

- Highlight the main problem regarding the tiny pollution water in Kuitun river.

- Write full name of the abbreviations.

- Include details on methodology (concentration of PAN, dosage of PAC, operating conditions of UF process)

- Conclude the abstract with future remarks.

Introduction:

-  Why Kuitun River is a slightly polluted river in Northern Xinjiang? Explain the pollution sources.

- The authors mentioned that the water in the Kuitun river is necessary to recycle it after treatment. What sort of treatment did the river undergo? Requires recycling or reuse?

- No pre-existing findings were discussed to demonstrate the importance of this study. Why the authors decided to use PAN as the UF membrane? What type of activated carbon used in this study? Please be specific.

- The novelty of this study should be clearly mentioned in the introduction part. The authors must present the research gap and the intention of this study to the readers. As far as I know, there is no significant innovation in using a UF membrane while observing the effect of the PAC dosage inside the CSTR.

Materials and Methods:

- The materials used in this study should be explained in detail, particularly in the case of commercial products.

- The supplier of the materials must be included in the materials.

- The experimental methods are too simple and hard to comprehend.

- All parameter ranges are required to be justified.

Results and Discussion:

- Superficial discussion.

- The removal rates of COD, NH3-N and Fe in wastewater by PAC - HUM combined process can reach 62%, 32% and 90% respectively. UF membrane is normally used as a pre-treatment process in the water and wastewater treatment. The UF membrane can be effective in reducing turbidity and TSS from contaminants in the wastewater, but it is not possible to remove COD as high as 62%. Explain how the PAN UF commercial membrane can eliminate most of the Fe (heavy metal) in wastewater? Usually, a specific reaction is required to remove the heavy metal.

Author Response

Response to Reviewer 1 :

Reviewer 1 :Comments and Suggestions for Authors

There are several comments to be addressed by the authors:

Abstract:

- Highlight the main problem regarding the tiny pollution water in Kuitun river.

- Write full name of the abbreviations.

- Include details on methodology (concentration of PAN, dosage of PAC, operating conditions of UF process)

- Conclude the abstract with future remarks.

Response: Thank you very much for your valuable advice. We have made detailed modifications according to your advices. Please see the red fonts in this part.

Introduction:

-  Why Kuitun River is a slightly polluted river in Northern Xinjiang? Explain the pollution sources.

- The authors mentioned that the water in the Kuitun river is necessary to recycle it after treatment. What sort of treatment did the river undergo? Requires recycling or reuse?

- No pre-existing findings were discussed to demonstrate the importance of this study. Why the authors decided to use PAN as the UF membrane? What type of activated carbon used in this study? Please be specific.

- The novelty of this study should be clearly mentioned in the introduction part. The authors must present the research gap and the intention of this study to the readers. As far as I know, there is no significant innovation in using a UF membrane while observing the effect of the PAC dosage inside the CSTR.

Response: Thank you very much for your valuable advice. We have made detailed modifications according to your advices. In the introduction, we have added the basic information, risk expectation and research progress of the HUM and HUM-PAC in the world. And at the middle of the introduction, we also added the research value and significance of this study to the HUM and HUM-PAC research in Xinjiang of China.

Xinjiang is an arid area with serious water shortage and backward water treatment and utilization technology. Kuitun city is an overexploited area of groundwater, and water resources are extremely scarce. Kuitun River is a slightly polluted river in Northern Xinjiang. It is necessary to recycle it after treatment. The water of Kuitun River will be used for irrigation and recycling after treatment.

Xinjiang is an underdeveloped area in China with limited economic conditions. The development of water treatment technology is still relatively preliminary. The methods used in Kuitun area are mainly traditional oxidation ditch and simple wetland treatment methods. The treated water quality cannot meet the requirements of local water resources recycling, and the membrane method has not been applied locally due to various restrictions. Therefore, it is very necessary to popularize and apply membrane for water treatment and membrane technology in Xinjiang. This study can lay a foundation for the efficient utilization of water resources and the development of new water treatment technologies in Xinjiang

Although there are many applications and researches of ultrafiltration and PAC, they have not been promoted and applied in many poor areas in China. Especially in the arid areas in northwest China, such as Xinjiang, we applied PAC-HUM for the first time. Kuitun River is a major river in northern Xinjiang, and large areas of farmland need to be irrigated by river water. The current traditional water treatment technology and simple constructed wetland technology can not treat water to irrigation and discharge standards. Due to economic constraints, the use of cheap membrane materials is more suitable for local conditions. We use hollow fiber ultrafiltration membrane, cheaper PAN material and locally available PAC for research.

We have added a lot of references, cases and explanations to the introduction.Please see the red fonts in this part.

Materials and Methods:

- The materials used in this study should be explained in detail, particularly in the case of commercial products.

- The supplier of the materials must be included in the materials.

- The experimental methods are too simple and hard to comprehend.

- All parameter ranges are required to be justified.

Response: Thank you very much for your valuable advice. We have made detailed modifications according to your advices. Please see the red fonts in this part.

Results and Discussion:

- Superficial discussion.

- The removal rates of COD, NH3-N and Fe in wastewater by PAC - HUM combined process can reach 62%, 32% and 90% respectively. UF membrane is normally used as a pre-treatment process in the water and wastewater treatment. The UF membrane can be effective in reducing turbidity and TSS from contaminants in the wastewater, but it is not possible to remove COD as high as 62%. Explain how the PAN UF commercial membrane can eliminate most of the Fe (heavy metal) in wastewater? Usually, a specific reaction is required to remove the heavy metal.

Response: Thank you very much for your valuable advice. We have made detailed modifications according to your advices.

  1. We have refined the discussion part and added many contents and references. And carried on the detailed explanation.
  2. UF membrane is generally used for advanced treatment of unconventional water. Generally, pretreatment is added for UF to achieve better treatment effect. In this study, we need to treat the water of Kuitun River to the standard that can be irrigated, and do not need to treat it to the standard of drinking water.
  3. The removal of these substances is mainly related to the physical interception of membrane pore size. It is also related to the increase of compaction layer formed by PAC and pollutants.The ferric ion removal rate of the PAC-HUM combined process was between 73% - 94.7%, which was significantly better than that of the single HUM. This is because with the increase of operation time and hydraulic retention time, PAC will form a compacted layer on the membrane surface after absorbing various pollutants. The interface between the compacted layer and the membrane will also form a concentration polarization layer due to pressure. In this way, the pore diameter of the membrane is equivalent to continuously reducing, removing the substances whose ionic radius is smaller than the membrane pore, and the removal efficiency is getting higher and higher. The radius of heavy metal ions will be removed in the process of operation compared with the pore diameter of the membrane which is constantly decreasing. However, with the increase of the compacted layer and concentration polarization layer, the flux will also decrease and the filtration pressure will increase, which requires us to clean and restore the membrane regularly.

Please see the red fonts in this part.

Reviewer 2 Report

1.         The introduction is too weak. It should be supported with relevant and recent references. The results of combining the adsorption process with the ultrafiltration membrane should be explained deeply.

 2.         Page 1, line 40 “due to its characteristics of high molecular weight cutoff…..” needs references.

 3.         The novelty of your work should be highlighted.

 4.         The environmental quality standard for surface water (GB3838-2002) should be added to the manuscript for better comparison with the effluent.  

 5.         The limitation of the present work should be mentioned for further studies.

 6.         The discussion is too shallow. The results should be compared with other works.

 7.         The legend of Fig.11 should be corrected.

 8.         How did you determine the period of backwashing? In which membrane flux you started to clean the membrane. 

Author Response

Response to Reviewer 2 :

Reviewer2 : Comments and Suggestions for Authors

  1. The introduction is too weak. It should be supported with relevant and recent references. The results of combining the adsorption process with the ultrafiltration membrane should be explained deeply.

Response: Thank you very much for your valuable advice. We have made detailed modifications according to your advices. In the introduction, we have added the basic information, risk expectation and research progress of the HUM and HUM-PAC in the world. And at the middle of the introduction, we also added the research value and significance of this study to the HUM and HUM-PAC research in Xinjiang of China. And we have added a lot of references, cases and explanations to the introduction. Please see the red fonts in this part.

  1. Page 1, line 40 “due to its characteristics of high molecular weight cutoff…..” needs references.

Response: Thank you very much for your valuable reminder. We have added the references.

  1. The novelty of your work should be highlighted.

Response: Thank you very much for your valuable advice. we have added the basic information, risk expectation and research progress of the HUM and HUM-PAC in the world. And at the middle of the introduction, we also added the research value and significance of this study to the HUM and HUM-PAC research in Xinjiang of China.

Xinjiang is an arid area with serious water shortage and backward water treatment and utilization technology. Kuitun city is an overexploited area of groundwater, and water resources are extremely scarce. Kuitun River is a slightly polluted river in Northern Xinjiang. It is necessary to recycle it after treatment. The water of Kuitun River will be used for irrigation and recycling after treatment.

Xinjiang is an underdeveloped area in China with limited economic conditions. The development of water treatment technology is still relatively preliminary. The methods used in Kuitun area are mainly traditional oxidation ditch and simple wetland treatment methods. The treated water quality cannot meet the requirements of local water resources recycling, and the membrane method has not been applied locally due to various restrictions. Therefore, it is very necessary to popularize and apply membrane for water treatment and membrane technology in Xinjiang. This study can lay a foundation for the efficient utilization of water resources and the development of new water treatment technologies in Xinjiang

Although there are many applications and researches of ultrafiltration and PAC, they have not been promoted and applied in many poor areas in China. Especially in the arid areas in northwest China, such as Xinjiang, we applied PAC-HUM for the first time. Kuitun River is a major river in northern Xinjiang, and large areas of farmland need to be irrigated by river water. The current traditional water treatment technology and simple constructed wetland technology can not treat water to irrigation and discharge standards. Due to economic constraints, the use of cheap membrane materials is more suitable for local conditions. We use hollow fiber ultrafiltration membrane, cheaper PAN material and locally available PAC for research.

We have added a lot of references, cases and explanations to the introduction. Please see the red fonts in this part.

  1. The environmental quality standard for surface water (GB3838-2002) should be added to the manuscript for better comparison with the effluent.  

Response: Thank you very much for your valuable advice. We have added the content and explanation of this part. Please see the red fonts.

  1. The limitation of the present work should be mentioned for further studies.

Response: Thank you very much for your valuable advice. We have added references, cases and explanations in the introduction.

  1. The discussion is too shallow. The results should be compared with other works.

Response: Thank you very much for your valuable advice. We have made substantial modifications according to your requirements, added a large number of explanations and references, and explained the principle in detail. Please see the red fonts.

  1. The legend of Fig.11 should be corrected.

Response: Thank you very much for your valuable reminder. We have corrected it.

  1. How did you determine the period of backwashing? In which membrane flux you started to clean the membrane. 

 Response: Thank you very much for your valuable advice. We have made substantial modifications according to your requirements, added a large number of explanations and explained the principle in detail.

With the increase of operation time and hydraulic retention time, PAC will form a compacted layer on the membrane surface after adsorbing pollution matters. The interface between the compacted layer and the membrane will also form a concentration polarization layer due to pressure. In this way, the pore diameter of the membrane is equivalent to continuously reducing, removing the substances whose ionic radius is smaller than the membrane pore, and the removal efficiency is getting higher and higher. However, with the increase of the compacted layer and concentration polarization layer, the flux will also decrease and the filtration pressure will increase, which requires us to clean and restore the membrane regularly

Normal operating pressure of ultrafiltration equipment is 0.06-0.12MPa (instrument display of control cabinet is 60-120). When the operating pressure of the ultrafiltration equipment reaches 0.12MPa (the instrument in the control cabinet displays 120), forced dosing backwashing is required.

Please see the red fonts.

Round 2

Reviewer 1 Report

The present paper investigates the "Study on Treatment of Tiny Pollution Water with PAC-HUM System in Kuitun River". The revised manuscript showed an improvement in the introduction and discussion parts. However, the revised manuscript is poorly written and contains many errors, and the findings were not discussed in detail.  The authors have not quoted accurately in the text and the novelty is not significant. In its present form, this article is not suitable for publication in Membranes.

Reviewer 2 Report

The applied changes are appropriate and the quality of the article has improved appreciably in new version.